# OpenReview forum: "Learning Compact Representations via Intrinsic Dimension Regularization"
_ICLR.cc/2026/Workshop/GRaM — ICLR 2026 Workshop GRaM Poster_

### Official Review · Reviewer_MKaf · 2026-02-17
**Promising geometric insight, but under-explored and flawed by weak baselines**

**Rating:** 5
**Confidence:** 4

**Review:**

The paper presents a novel regularization scheme that penalizes the effective rank of the learned representations. The main contribution is to introduce the IDDR loss that defines the "Goldilocks zone" and nudges the soft effective rank of the representations to lie within it. Using the effective rank, the authors derive a generalization bound for classification that scales with effective rank and thus shows that lower effective rank yields a tighter bound.

**Strengths and weaknesses**

The main strength of the paper is to introduce as a differentiable, scalable way to penalize the rank of representations globally (wherever we have data, that is). Another strength is the derived generalization bound that scales with the effective rank.

The main weakness of the paper is that it focuses only on classification, whereas the method could be applicable for generation, transfer learning, OOD detection. Not only can it be applicable in these domains, but it is not clear that this sort of penalty is good for those tasks. There is a question to be asked whether the benefits are classification specific or they truly reflect the properties of the underlying data manifold.

Another weakness are the experiments that show a mixed picture, where the models generally have bad performance (both MLP and CNN based) and IDRR regularization often relies on Dropout to be competitive.

**Experiments**

- The CNN baseline on CIFAR achieves only 68.8% accuracy, whereas standard simple CNNs easily exceed 80% and ResNets exceed 90% on this dataset. Showing a marginal improvement (+0.5%) on a fundamentally weak/under-performing backbone does not demonstrate utility.

- The method heavily relies on being combined with Dropout (IDRR+Dropout) to match baseline performance. In isolation, IDRR often underperforms even the unregularized Standard baseline (e.g., CIFAR-10 MLP: 44.9% vs 46.0%). Note, that Dropout can be seen as a technique that "spreads" the information around, avoiding to rely on specific dimensions due to random connections disappearing. In this sense, it acts counter to the IDRR regularization. This suggests the geometric constraint is too aggressive and requires Dropout's robustness to function, contradicting the claim that IDRR is the primary driver of performance.

- The comparisons with Weight Decay and Dropout are slightly misplaced. These regularization methods operate via fundamentally different mechanisms (weight norm penalties vs. stochastic noise), making direct comparison with a spectral geometric constraint less meaningful than comparisons with other geometric regularizers or bottleneck methods.

Furthermore, the paper claims to produce compact representations, but this is strictly spectral, not computational. The ambient dimension ($D$) and the weight matrices remain unchanged. Thus, the method offers no inference-time savings in memory or compute. The "compression" is theoretical, not practical, unless accompanied by an explicit pruning step which is absent.

I am also curious about Figure 1 and how to understand the PCA projections as there appears to be a significant discrepancy between the quantitative metrics and the visual analysis.Table 1 claims an Effective Rank of 12.6 for MNIST (IDRR+Dropout) while Figure 1 (Top Right)  shows the data collapsing almost entirely onto a single horizontal stripe (PC2), with the 9 class clusters bunched in intervals along this line. If the effective rank were truly ~12.6, there should be significant variance across at least 12 dimensions. A projection that looks like a 1D line implies an effective rank closer to 1. This suggests either the metric is inflated or the representation has collapsed to a brittle 1D manifold, which is concerning for robustness and use cases beyond classification.

**Pmlr Suitability:**

Yes

---

### Official Review · Reviewer_Q6st · 2026-02-24
**Learning Compact Representations via Intrinsic Dimension Regularization**

**Rating:** 5
**Confidence:** 2

**Review:**

This paper proposes IDRR, a training-time regularizer that controls representation dimensionality by penalizing the soft effective rank (entropy of normalized singular values) of batch activations. Empirically, IDRR+Dropout matches dropout accuracy while reducing effective rank by 68-81% on several small-scale benchmarks, with supporting geometry visualizations of sharper singular-value decay.


### Strengths
- The paper provides good theoretical motivation connecting effective dimension to generalization bounds.
- The regularizer is simple, differentiable, and directly targets a measurable notion of representation compactness (effective rank).
- Empirically, the rank reductions are large and consistently reported across MLP and CNN settings.


### Weaknesses
- The theory assumes a low-dimensional manifold and links bounds to a dimension parameter d, but the operational quantity being regularized is batch effective rank, so the bridge from practice to theory remains somewhat indirect.
- The computational cost of per-batch SVD is acknowledged but not quantified. Scalability to larger widths networks remains untested.
- The empirical setting is limited (e.g., MLP with low accuracy on flattened CIFAR-10), so it is unclear whether results hold in stronger modern training regimes.

**Pmlr Suitability:**

Yes

---

### Official Review · Reviewer_zs96 · 2026-02-24
**Review for Learning Compact Representations via Intrinsic Dimension Regularization**

**Rating:** 4
**Confidence:** 3

**Review:**

The paper proposes Intrinsic Dimension Regularization for Representation Learning (IDRR), which explicitly controls representation dimensionality through differentiable regularization.

Even though the idea is interesting, I have some major concerns:
1. The performance drops in many cases when used without Dropout
2. Even with Dropout (i.e., IDRR + DropOut) is minimal in most cases over simply using Dropout
3. It would have been interesting to see the performance improvement in larger datasets on standard CNNs

As such, what is the utility of IDRR if a simple Dropout performs better?

**Pmlr Suitability:**

Yes

---

### Meta-Review · Area_Chair_Kpaj · 2026-02-24

**Decision:**

Accept

**Metareview:**

The authors present Intrinsic Dimension Regularization for Representation Learning (IDRR) that penalizes the effective rank in a differentiable way to improve accuracy when combined with dropout. All the reviewers appreciated the results, found it interesting and relevant, while having some concerns. I strongly suggest that the authors incorporate the reviewers' comments into the next version of their paper.

**Relevance To Proceedings:**

Yes — suitable for PMLR (long paper)

**Relevance To Workshop:**

Yes — suitable for GRaM

---

### Decision · Program_Chairs · 2026-03-02

Accept (Poster)